A new approach of anomaly detection in shopping center surveillance videos for theft prevention based on RLCNN model

http://orcid.org/0000-0003-4896-2025 Sajid Muhammad 1 msajid@aumc.edu.pk
Khan Ali Haider 2 3
http://orcid.org/0000-0003-4538-3095 Malik Kaleem Razzaq 1
http://orcid.org/0000-0003-3306-1195 Khan Javed Ali 4
Alwadain Ayed 5 aalwadain@ksu.edu.sa
1 Department of Computer Science, Air University , Islamabad , Pakistan
2 Department of Software Engineering, Faculty of Computer Science, Lahore Garrison University , Lahore, Punjab , Pakistan
3 College of Computer Science, Beijing University of Technology , Beijing , China
4 Cybersecurity and Computing Systems Research Group, Department of Computer Science, University of Hertfordshire , Hatfield , UK
5 Computer Science Department, Community College, King Saud University , Riyadh , Saudi Arabia
Benítez-Andrades José Alberto
Electronic publication date: 2025 Jun 18
Publication date: 2025
Volume: 11
Electronic Location ID: e2944
Received 2025 Feb 18; Accepted 2025 May 16
Copyright: © 2025 Sajid et al.
Copyright year: 2025
Copyright holder: Sajid et al.
License: This is an open access article distributed under the terms of the Creative Commons Attribution License, which permits unrestricted use, distribution, reproduction and adaptation in any medium and for any purpose provided that it is properly attributed. For attribution, the original author(s), title, publication source (PeerJ Computer Science) and either DOI or URL of the article must be cited.
License URL: https://creativecommons.org/licenses/by/4.0/

Keywords: Human activities, CNN-LSTM, Surveillance, Anomaly detection, UCF datasets, RLCNN

Funding: King Saud University, Riyadh, Saudi Arabia ORF-2025-309 This research is supported by Ongoing Research Funding program, (ORF-2025-309), King Saud University, Riyadh, Saudi Arabia. The funders had no role in study design, data collection and analysis, decision to publish, or preparation of the manuscript.

==============================
The amount of video data produced daily by today’s surveillance systems is enormous, making analysis difficult for computer vision specialists. It is challenging to continuously search these massive video streams for unexpected accidents because they occur seldom and have little chance of being observed. Contrarily, deep learning-based anomaly detection decreases the need for human labor and has comparably trustworthy decision-making capabilities, hence promoting public safety. In this article, we introduce a system for efficient anomaly detection that can function in surveillance networks with a modest level of complexity. The proposed method starts by obtaining spatiotemporal features from a group of frames. The multi-layer extended short-term memory model can precisely identify continuing unusual activity in complicated video scenarios of a busy shopping mall once we transmit the in-depth features extracted. We conducted in-depth tests on numerous benchmark datasets for anomaly detection to confirm the proposed framework’s functionality in challenging surveillance scenarios. Compared to state-of-the-art techniques, our datasets, UCF50, UCF101, UCFYouTube, and UCFCustomized, provided better training and increased accuracy. Our model was trained for more classes than usual, and when the proposed model, RLCNN, was tested for those classes, the results were encouraging. All of our datasets worked admirably. However, when we used the UCFCustomized and UCFYouTube datasets compared to other UCF datasets, we achieved greater accuracy of 96 and 97, respectively.

Introduction

Losses in revenue, interruptions in operations, and morale among employees can all stem from retail fraud and burglary. Executives in the retail sector are understandably worried about these unethical practices because they threaten the long-term health of their companies, their employees, and the economy as a whole. The purpose of this practical empirical study was to examine the strategies used by leaders to prevent theft and thievery, drawing from the theoretical framework of retail (Barcon, 2025; Sivalakshmi et al., 2024). The need to ensure public safety while reducing casualties during events with religious or civic significance has led to a rise in the use of automated crowd assessment systems in recent years. However, it is still difficult to accurately evaluate the imperfections in real-life crowd photos and videos (Garg et al., 2025; Xu et al., 2025). To provide safety and oversight in many sectors, including public spaces and private organizations, camera recording devices are required. The cameras in these systems capture and record video footage, allowing for monitoring and analysis in real-time. There has been a growing need for video-based solutions to enhance the functionality and performance of security systems due to the proliferation of the Internet of Things (IoT) (Xu & Zhai, 2024). According to Qiu et al. (2022), Tahir et al. (2022), human activity recognition aims to detect the physical movements of both individuals and groups. The processing speed, size, accuracy, and manufacturing costs of sensors have all seen considerable advancements in the last 10 years (Liu et al., 2016).

Thanks to these innovations, a wide variety of sensors may be integrated into smartphones and other mobile devices, making them smarter and more useful. There are several benefits to using closed-circuit television (CCTV) or video surveillance technology, such as lower expenses, better video quality, easier installation, and increased security (Bux, Angelov & Habib, 2017; Dang et al., 2019). According to Jagadeesh & Patil (2016), CCTV systems are thus being used in numerous monitoring and security applications. Although different kinds of sensors are made for specific tasks, most sensors acquire raw data from a wide variety of sources and then analyze it to draw broad conclusions. In recent times, there has been an uptick in interest in human activity recognition (HAR) which involves studying human movement using wearable sensors. HAR—the automatic identification of physical movements—is being studied and used more and more in human-computer interface (HCI), mobile computing, and pervasive computing. Computing frameworks can more effectively assist users in their attempts with the data provided by HAR (Abowd et al., 1998). To make our communities safer, more and more places are installing surveillance cameras in high-traffic locations like roads, junctions, banks, and shopping malls (Rahim et al., 2023). The health and happiness of the elderly are directly related, making it all the more important for society to emphasize wellness (Sajid et al., 2022). Law enforcement agencies’ surveillance capabilities have failed to keep pace. As a result, there is an unacceptable ratio of surveillance cameras to human monitors and a large variance in their distribution.

Unusual events, such as accidents, criminal activity, or illegal conduct, must be detected by video monitoring. Usually, anomalies appear far less frequently than normal activity. Innovative computer vision algorithms for automatic video anomaly detection are desperately needed to save time and effort (Yasin et al., 2023). Finding activities that deviate from expected patterns and determining the most likely timing of the anomaly are the goals of a successful anomaly detection system. A basic method for understanding videos is anomaly detection, which distinguishes between expected patterns and oddities. When a monster is found, it may be assigned to a certain task using classification techniques. The first step in addressing anomaly detection is developing algorithms to identify particular anomalous occurrences, such as a violence detector (Mohammadi et al., 2016) or a traffic accident detector (Sultani & Choi, 2010). These techniques, however, are insufficient to detect additional anomalous occurrences. As a result, their usefulness is completely limited. Anomalies in the real world take many complex forms. It is difficult to list every possible aberrant event. To lessen this problem, the anomaly identification algorithm must not rely on past knowledge of events (Ahmed et al., 2023). As a result, anomaly detection should be carried out with little supervision. Sparse-coding techniques (Lu, Shi & Jia, 2013) are great tactics that produce anomaly identification outcomes that are at the forefront of the field. The basic idea behind anomaly detection is that irregular events cannot be precisely reproduced by utilizing the language of regular events. However, the environment that security cameras record might change significantly over time (for instance, during different times of the day), which can result in higher false alarm rates for a variety of routine activities (Muneer et al., 2022; Sana et al., 2022; Gao et al., 2016).

Significant progress has been made in the area of object identification and monitoring in IoT-effective retail monitoring systems in recent years (Xu & Zhai, 2024). The development of accurate and effective techniques for real-time object tracking and detection has been the focus of numerous investigations. Traditional approaches relied on manually created features and rule-based algorithms, which lacked stability and occasionally struggled in complex scenarios. Significant improvements in object detection have been made possible by the development of deep learning techniques (Sajid et al., 2025b), particularly convolutional neural networks (CNNs) (Xu & Zhai, 2024). A novel benchmark dataset designed especially for the early detection of suspicious behavior that may lead to theft is presented in a recent study (Shrestha, Taniguchi & Tanaka, 2024). The relevant sequences from the UCF101-Crime dataset, such as suspicious activity, actual shoplifting incidents, and typical behaviors, were selected and segmented to create this dataset (Sharma & Kanwal, 2024). The dataset consists of short clips, for which we suggest a robust neural network architecture that combines the benefits of bidirectional long short-term memory (BiLSTM) networks for capturing temporal dependencies in action sequences and convolutional neural networks (CNN) for efficient feature extraction (Shrestha, Taniguchi & Tanaka, 2024).

CNN is used in our suggested model to extract features. To handle the video data, we use convolutional long short-term memory (ConvLSTM) and a recurrent neural network (RNN) in our modeling frameworks. The model then determines whether the given video has any prohibited content. This approach saves money and human resources while improving the accuracy of identifying irreparable losses. The suggested approach can drastically cut emergency service response times, which is crucial for the public and governments (Moreira et al., 2021). To identify anomalous occurrences in our time-series data, this study developed a model that first employed a CNN to extract significant information from each input sequence frame. Next, it employed a unique recurrent neural network, followed by an LSTM, and a CNN (RLCNN) structure. Additionally, we categorize both typical and unusual actions, demonstrating the system’s ability to determine the appropriate classification for every anomaly. Our suggested RLCNN method’s main contributions are as follows: This work established a human activity detection system aimed at detecting aberrant behaviors associated with theft in video feeds from security surveillance cameras in shopping malls, hence enhancing security and mitigating losses.

In contrast to cutting-edge methodologies for human activity prediction, our model is designed to identify a broader spectrum of actions, including stealing and the handling of suspicious objects, hence facilitating precise detection of anomalous human behavior.

To enhance the precision of anomaly detection, we developed a tailored dataset by amalgamating UCFYouTube and UCF101, which will facilitate training data and enable the model to discern unique patterns linked to aberrant activity.

This study utilized many datasets to train the model for effective anomaly detection, aiming to surpass the performance of contemporary methods.

The RLCNN model we proposed is good at finding unusual patterns in the UCF benchmark dataset while using fewer parameters and being smaller than earlier models. This improves the reliability and efficiency of our methodology, applicable to real-time surveillance in retail malls.

Problem statement

The key to turning indirect monitoring into a proactive practice is creating a system that can accurately predict and classify customers’ normal or unusual daily actions in shopping video surveillance. To identify typical purchasing behaviors, identify interactions, identify abnormal activities like theft or antagonism, and predict future actions, a system that goes beyond simple item tracking is needed. This can be accomplished by developing a system that improves security by identifying risks in real time and preventing losses, which would raise customer satisfaction. The results of the model can generate data-driven suggestions for crucial decision-making, automate analysis, provide actionable insights, and improve safety through accelerated instant action. To stop theft or other unlawful activity in a crowded shopping center, deep learning algorithms can be used to collect crucial data on consumer behavior. Creating a model for businesses in underdeveloped, low-income countries will decrease stealing and theft, which could boost retail industry profitability.

Organization

This article is structured as follows: Background offers a comprehensive analysis of the background. The methodology defines the research technique employed in this study. Results and Discussion delineate the findings and interact in discourse over these outcomes. The conclusion synthesizes the study’s principal implications.

Background

In quasi-data streams, the trained model over the primary data cannot be considered sufficient if fluctuations in the incoming data are detected. The adaptation challenge stems from the new data distribution, which demands variety in a quasi-environment (Lobo et al., 2018). To get over this problem, (Lobo et al., 2017) employs a bio-inspired algorithm to assess drift heterogeneity and obtain a broad range using a self-learning optimization technique. Another novel approach is the adaptation and enhancement of weighted one-class SVM to take quasi-streaming data into account (Krawczyk & Woźniak, 2015). They asserted that to take into account new data streams, a one-class classifier might modify its decision threshold. A forgetting mechanism allows the model to relearn its parameters. The introduction of an efficient ensemble learning technique for real-time activity recognition in this (Krawczyk, 2017) is comparable. Even in the absence of an external drift detector, the system iteratively adjusts the weights of the Naive Bayes classifier to make it smoothly adaptive to the stream’s condition. A comprehensive study on activity recognition in online data stream mining was carried out, according to Abdallah et al. (2018). Accurately identifying human motions in real-time from an online surveillance data stream is challenging due to a variety of factors, including the computation of high-dimensional characteristics, perspective variation, activity, and crowded backgrounds (Chang et al., 2016). Several handcrafted local feature descriptors have been used to recognize human behaviors to address these challenges during the past 10 years (Scovanner, Ali & Shah, 2007), when there were several spatial-temporal-based strategies. Bag-of-Words (Peng et al., 2016) can be used to improve the performance of these algorithms, which are based on motion data analysis (Marszalek, 2008). Nevertheless, Bag-of-Words development requires rigorous engineering and is computationally costly.

In a study (Shi, Laganiere & Petriu, 2015), 3D gradient orientation obtained from integral video representation is binned into a polyhedron to analyze appearance and motion data. This method has a significant quantization cost because of its high dimensionality. To solve this problem, (Caetano, dos Santos & Schwartz, 2016) suggested gradient boundary histograms. They used time derivatives of the image gradients to show the moving edge boundaries rather than simple gradients. Poulose, Kim & Han (2022) suggests a human image threshing (HIT) machine-based HAR system that employs an image dataset from a smartphone camera for activity recognition to overcome some of the current limitations of HAR systems. The HIT machine employs a deep learning model for activity classification, a facial image threshing machine (FIT) for image cropping and resizing, and a mask region-based convolutional neural network (R-CNN) for human body detection. Muhammad, Ahmad & Baik (2018) created the optical flow co-occurrence metric to derive a set of statistical measures based on the magnitude and direction of optical flow. The main motivation for the creation of the optical flow co-occurrence matrix was the assumption that the spatial connection, which greatly facilitates the description of motions, is present in the immediate neighborhoods of the flow field. Handcrafted feature extraction procedures, which have a high degree of extraction and classification complexity and convey low-level semantics in visual data, employ complicated engineering. Therefore, automatic feature learning methodologies are initiated by researchers. Neural network-based methods, for example, can directly extract features from raw inputs based on training weights and biases.

Unsupervised anomaly detection techniques, such as Cao et al. (2022), exclusively learn from normal data within designated categories. Unsupervised anomaly detection approaches often model the distributions of normal samples during training and then evaluate test samples against the established normal sample distribution to identify abnormalities. An effective method involves collecting features from each sample with pre-trained neural networks (Cai et al., 2023), followed by modeling the feature distribution through knowledge distillation (Liu et al., 2024; Sajid et al., 2024) and reconstruction (Sajid et al., 2025c; He et al., 2024). Semi-supervised anomaly detection techniques, such as Cao et al. (2024), necessitate annotated typical and unusual images from targeted classes for training purposes. They generally employ annotated atypical samples to develop a more concise delineation boundary for normal samples. Due to the utilization of supplementary aberrant samples, they generally exhibit superior anomaly detection performance relative to unsupervised anomaly detection (Barcon, 2025; Garg et al., 2025), although they impose stringent data requirements (Xu et al., 2025). Although conventional anomaly detection algorithms demonstrate promising efficiency, their efficacy tends to decline when normal samples are scarce for training. A study (Garg et al., 2025) introduces an innovative method for analyzing human crowd behavior by combining segmentation and classification using deep learning architectures (Pavithra et al., 2024). In contrast to current methodologies (Zahra et al., 2024), the study suggested an approach that employs an expectation–maximization-based ZFNet architecture for video scene segmentation, facilitating enhanced precision in the delineation of crowd dynamics. Furthermore, the study presents transfer exponential conjugate gradient neural networks for classification, improving the accuracy of crowd behavior characterization.

The method that CNN learns features is hierarchical, with the last layers extracting global features that represent high-level semantics from the first layers’ local parts they know from visual data (Xiao et al., 2023). Various temporal information fusion techniques are investigated to fuse local motion direction with global features. However, the 63.3% (Ahmad et al., 2018) identification rate for the UCF 101 dataset demonstrates that their neural architecture cannot effectively capture human behaviors in the video stream. The activation of pre-trained CNN models demonstrated excellent success for image retrieval, fire detection, and video summarizing, according to the findings of numerous studies (Casagrande, Tørresen & Zouganeli, 2019). As a result, we have thoroughly examined the profound properties of numerous CNN models trained to recognize actions. Moreover, the cost of computing using current CNN models is high, and their identification accuracy does not satisfy all contexts, for example, quasi- or actual data streams. For persons with dementia or cognitive impairment, sequence prediction gives them the ability to anticipate in advance what will happen next (Casagrande, Tørresen & Zouganeli, 2019). The authors in Jebur et al. (2022, 2023) discussed that hybrid models amalgamate multiple frameworks to optimize adaptive delivery for video streaming. These models are also proficient with high-dimensional input data, including video data. Deep hybrid models mostly utilize deep neural networks for feature extraction and traditional machine learning algorithms to detect anomalous activities. CNNs are powerful deep learning algorithms adept at collecting salient features from video frames and categorizing them based on their content. Compared to earlier works, we adopted a new approach. We applied sequence prediction algorithms to a series of actions already identified by an activity recognition classifier and categorized as routine by an anomaly detector. Then, LSTM one-hot encoding is applied to a sequence of operations. As this technique does not require changing the dataset, we used it in our research to identify anomalous instances. A comparative analysis of the existing techniques in recognition of human behavior is given in Table 1.

Table 1 Comparative analysis of existing techniques.

Ref	Purpose	Method	Key points	
Krawczyk (2017)	Activity recognition	Ensemble learning	Presents effective ensemble learning	
Abdallah et al. (2018)	Activity recognition	Data stream mining	Activities recognized	
Scovanner, Ali & Shah (2007)	Human actions	Spatial-temporal-based techniques	Handcrafted local feature descriptors	
Shi, Laganiere & Petriu (2015)	Appearance and motion data	Binning 3D gradient orientation	Quantization cost & high dimensionality	
Poulose, Kim & Han (2022)	HAR system	Human image threshing (HIT) machine	Use image dataset from smartphone camera	
Cao et al. (2022)	Unsupervised Anomaly Detection	Learn from normal data	Learn exclusively	
Cao et al. (2024)	Semi-supervised anomaly detection	Annotated images	Requires annotated images	
Garg et al. (2025)	Analyze human crowd behavior	Eegmentation & classification	Deep learning architectures	
Xiao et al. (2023)	Anomaly Detection	Extracting features hierarchically	Last layers extract global features	
Casagrande, Tørresen & Zouganeli (2019)	Sequence prediction	LSTM	Next sentence prediction	
[Ours]	Human behavior recognition	RNN-LSTM-CNN	Predicted human activities	

Methodology

The RLCNN model is adept at anomaly detection in shopping center surveillance films, as it proficiently captures the video data’s temporal and spatial dimensions. Surveillance recordings are intrinsically sequential, depicting events that occur across time. The recurrent connections in the RLCNN let it retain knowledge from previous frames, facilitating comprehension of the present frame and forecasting future events.

This is essential for identifying anomalies that entail behavioral changes over time, such as loitering or hiding goods. Moreover, the convolutional layers of the RLCNN effectively extract local spatial data from each frame, including object movements and interactions. The RLCNN synergizes the advantages of RNNs and CNNs in proficiently discerning intricate patterns of typical activity and detecting anomalies that may signify theft. Convolutional neural networks are impressive when dealing with image data and the problem of image classification, while LSTM methods are great when dealing with data sequences. A general flow of the human activity recognition is shown in Fig. 1.

Figure 1 General flow diagram for human activity recognition.

Preprocessing

Data mining and machine learning techniques necessitate pre-processing, as the applicability and quality of the data profoundly influence the outcomes and effectiveness of the models. Images frequently display noise owing to insufficient quality, requiring pre-processing. In a study (Sajid et al., 2025a, 2025b), the authors specified that the objectives during the preprocessing phase encompass image stabilization and the minimization and refinement of noise reduction. The authors emphasized the significance of preprocessing. Various important image features are shown in Fig. 2 that provide different techniques for image preprocessing. The authors indicated that the impact of image preprocessing, especially segmentation, on classification accuracy has not been extensively examined in various studies. Addressing these restrictions is crucial for the advancement of efficient and pragmatic anomaly detection methodologies. Preprocessing creates a consistent foundation for the augmentation procedure. Initially, we employed diverse methodologies to preprocess the data. Subsequently, we utilized augmentation approaches on the preprocessed data to avert information loss during the training phase. The integration of many augmentation techniques can enhance model performance and generalization. When selecting the most appropriate augmentation strategies, the characteristics of the anomalous images and the intended augmentation goals—such as improving model robustness, reducing overfitting, or expanding the training dataset—are taken into account (Sajid et al., 2025a).

Figure 2 Various important image features in recognition of human activity process.

The suggested study generates features and labels using a convolutional neural network feature extraction model, as explained in Fig. 3. CNN can extract video information from pooling, convolutional, and fully connected layers. CNN extracts the following numerous significant elements:

Figure 3 Different stages of CNN feature extraction in the shopping center areas.

Frame extraction: Continuous monitoring videos encompass a substantial volume of data. Processing each frame requires significant time and computer resources; thus, frame extraction facilitates the selection of a subset of frames, thereby minimizing processing overhead. Not every frame in a video is significant for specific activities. Extracting keyframes emphasizes the most informative segments of the video. The retrieved frames serve as input for feature extraction, which entails deriving significant properties to detect anomalies (Dang et al., 2020).

Resizing: Machine learning techniques for anomaly detection prediction necessitate input photos of uniform dimensions. It is crucial to ensure the uniformity of all data input into the model. High-resolution images require increased memory and processing power. Reducing images to a smaller, manageable size alleviates the computing load, thereby enhancing the speed and efficiency of anomaly identification. Resizing helps optimize images for effective feature extraction.

Normalization: Pixel values in images range from 0 to 255. Normalization adjusts these values to a more uniform range, typically between 0 and 1 or −1 and 1. This will guarantee that all input data maintains a comparable scale, preventing certain aspects from overshadowing others. It facilitates the accelerated and more efficient convergence of deep learning models during training. Normalizing pixel values enhances the quality of retrieved features, facilitating the ability of the anomaly detection model to discern subtle patterns and anomalies. Normalization enhances the numerical stability of computations executed by the anomaly detection algorithm (Gill et al., 2022).

Data augmentation: Data augmentation is a method employed to enlarge a training dataset artificially by generating altered versions of existing data. This is especially significant in machine learning, particularly deep learning because models frequently necessitate substantial quantities of data to function well. Overfitting occurs during model training when the model excessively learns the training data, including its noise, leading to subpar performance on unseen data. Data augmentation introduces variations that enhance the model’s generalization capabilities. Data augmentation enhances the model’s robustness to real-world settings by exposing it to a broader spectrum of data changes.

Data splitting: To assess the classifier’s effectiveness, it is typical to divide the dataset into training and testing sets.

Figure 4 provides a block diagram of RLCNN. Here, we take the input and perform preprocessing to prepare the dataset for RLCNN input. Finally, we assess our model and make predictions about human activity. We will combine ConvLSTM cells to implement the initial strategy in this stage. ConvLSTM cells are variants of the LSTM network that incorporate convolutional processes. It is an LSTM with convolution included in the architecture, enabling it to distinguish between spatial and temporal input elements (Arshi, Zhang & Strachan, 2019). This technique effectively captures individual frames’ temporal and geographic relationships to categorize videos. This convolution structure allows the ConvLSTM to receive input in three dimensions (width, height, and several channels). An LSTM cannot independently represent spatiotemporal data since a simple LSTM can only accept information in one dimension.

Figure 4 Proposed model RLCNN preprocessing images from videos, training, evaluation, and testing block diagram.

This particular LSTM only functions with 3D data, not 1D. Convoluted operations are ingrained inside it (Al-Kahtani et al., 2023). The model will be constructed using Keras ConvLSTM2D (Xavier, 2019) recurrent layers. The kernel size and different filters needed to apply convolutional operations are also considered by the ConvLSTM2D layer. The output of the layers is flattened and then given to the dense layer, which calculates the probability for each action category using Softmax activation. In addition, we will use dropout layers to stop the model from overfitting the data, MaxPooling3D layers to reduce the size of the frames and remove unnecessary calculations, and both. The straightforward design has few trainable parameters. This is because only a tiny percentage of the dataset is used, which does not necessitate a sophisticated model (Masud et al., 2023).

RLCNN architecture

The ConvLSTM2D model is enhanced with layers to create a cell for a specific network. A network has a greater capacity for learning as more filters are applied. More filters are introduced as one delves deeper into the network, including 2, 4, 8, 14, and 16. More and more feature maps are being added to the grid in the CNN image. Since there are more filters, the network has more features. Pooling layers are utilized to reduce the size of the feature map after each convolutional layer has been added, which speeds up processing. It enables the network to train more precisely and have a broader view. Maxpooling3D is utilized in the network because we are working with an array of image frames and an image sequence. Whenever pooling is used, the size is often cut in half. The network is expanded to four convolutional layers. Each time, max pooling is added with increasing filters to decrease the size of feature maps and improve prediction accuracy. A dropout layer is present in a different time-distributed layer. By employing a dropout layer, the network can rely on overall weights rather than specific consequences. We apply a dropout layer to each extracted image’s feature map as we work with a video dataset.

The use of the time-dispersed layer is due to this. It ensures that everything included within this layer applies to each channel. At the network’s end, all the layers are flattened. Combine all of the feature maps, then flatten the result. There is a dense layer at the network’s terminus. There are as many nodes in it as there are classes. As a result, a network trained on four categories will have four outputs in its dense layer. We must calculate probabilities since we are working with the Softmax activation function. So, all the classes would have equal probability 1. It distributes probabilities across all the nodes. The ConvLSTM model is then assembled and trained. When the network no longer reduces loss or improves accuracy, the early stopping idea is utilized to stop teaching the network. When the best loss and accuracy are attained, the weights are restored. After training, the model is assessed, and accuracy and loss are shown. The model is plotted to determine whether it is well-fitted, over-fitted, or accurately and successfully generates the loss.

Working of RLCNN model

Using features extracted by CNN, a straightforward Algorithm 2 has been developed to recognize human activity. Inputs are then sent to an LSTM to detect human activity.

For tasks combining sequential data (inputs or outputs), visual, linguistic, and deep hierarchical feature extractors (like a CNN), as well as other types of data, an RNN (Mansouri et al., 2022), LSTM (Sherstinsky, 2020), and CNN are combined into a single model (RLCNN). This model includes a model capable of identifying and combining temporal dynamics. By extracting and labeling videos based on the best characteristics of CNN and RNN, respectively, LSTM classifies the actions. A memory cell combines LSTM and RNN (Yan, Xiong & Lin, 2018) to train the suggested classifier.

We have employed Algorithm 2 to detect human behaviors, both normal and aberrant. The Algorithm 2 technique is delineated by using the database (db) as the number of frames (nf) derived from Algorithm 1. Initially, we store the database in a temporary variable (S) for subsequent processing. The variable (T) contains the temporal sequences of the frames collected in Algorithm 1 in time T seconds. Furthermore, N serves as our counter variable, initialized to 1. We traverse the video frames obtained through the temporary instance variable (Vi). We obtain the initial value (Vi) in a variable (T) to identify activities. We utilized ConvLSTM to extract frames at time T seconds, and the outcome is kept in the variable (Fr1). Subsequently, we input the ConvLSTM (Fr1) output into RLCNN to further analyze the video frames in temporal sequence for anomaly detection, with the results recorded in (Fr2).

Algorithm 1 Algorithm for feature extraction.

Require: nf≥20	
Ensure: Out=db	
  Step 1:	
  nf←1	
  db←0	
  V←totalframes	
  Step 2:	
  while V do	
     nf←frameN∗frameskip	
  Step 3:	
     if success then	
      N←nf	
      db=db+N	
  Step 4:	
     else	
       resizedframes←resizeframe	
       normalizedframes←resizedframe255	
       db=db+resizedframes+normalizedframes	
    end if	
  Step 5:	
     N←N+1	
  end while	

Algorithm 2 Algorithm for human activity recognition.

Require: db≥nf	
Ensure: Activity in time (T)	
  Step 1:	
  s←db	
 load model	
  T←time_in_seconds	
  N←1	
  Step 2:	
  while Vi do	
    T←Vi	
    Fr1←C1T	
    Fr2←C2Fr1	
    Prc←C2Fr2	
    Activity←Prc	
  Step 3:	
  if success then	
   Activity in time (T) ←db + N	
 Step 4:	
     else	
        Repeat:Prc←C2Fr2	
     end if	
  Step 5:	
     N←N+1	
  end while	

Subsequently, we iterate, utilizing the variable (Prc) counter to address the residual instances. In the concluding phase, activities are forecasted utilizing the (Prc) counter within T seconds. To understand Algorithm 2, we assess the success state. If activities are found, they are added to the database; otherwise, we return to Step 4 to identify another activity and increment the counter value. This approach enables the integration of CNN with LSTM to identify temporal dependencies, as well as the combination of CNN with recurrent neural networks to enhance model predictions. Utilizing Algorithm 2, we employed various parametric strategies that yielded optimal results for the anomaly detection process, with our model surpassing other state-of-the-art models.

(1) zmk=f(∑vNkXcmk+bmk)

where zmk is the feature map in the jth layer of the RLCNN, cmk shows the jth convolution filter, and Nk shows the set of feature maps. An RNN-like structure is the LSTM architecture. RNNs’ long-range dependencies and memory backup make LSTMs more accurate and efficient than conventional RNNs (Arifoglu & Bouchachia, 2017). LSTMs were developed to describe temporal sequences. The method is applied following data preprocessing, which removes unwanted, missing, and null signal values. The LSTM provides a solution by including a memory cell (C) to encode knowledge at each phase reliably. The memory cell (yt) is controlled by an input gate (rt),forgetgate (m t), and an output gate. These gates monitor the input that is read during categorization. These control gates also help the LSTM send the information to an unsupervised deep-learning hidden state without changing the output. The definition of the LSTM gates and updating at time t (Gill et al., 2022) is:

(2) rt=S(ZxrXt+Zhrht−1+ZvrVt+br)

(3) mt=σ(ZxmXt+Zhmht−1+ZvmVt+bm)

(4) nt=σ(ZxnXt+Zhnht−1+ZvnVt+bn)

(5) yt=tanh⁡(ZxcXt+Zhcht−1+ZvcVt+bc)

(6) Ct=mt⊙Ct−1+rt⊙yt

(7) ht=nt⊙tanh⁡Ct

where ⊙ shows multiplication operation, σ is the sigmoid function σ(x)=1(1+e−x) and tanh is hyperbolic tangent function tanh⁡(x)=ex−e−xex+e−x. Whereas rt,mt,nt,yt are the input gate, forget gate, output gate and input gate, respectively. x, h,v, and c are the input vector, hidden state, image feature, and memory cell. We use one input, a feature vector from the video frames we feed to LSTM, which only requires one block of data to train a series of captions instead of the video captioning model (Ullah et al., 2021).

(8) jt=1h∑t=1hZ¯itfi

(9) P¯t=Z¯Ttanh⁡(Z¯hht+MhRf+bh)

(10) Ct=softmax(P¯t)

where Z¯T,Z¯h,Mh,andbh are the learned parameters from frame features fi according to the weight Z¯it to return the score P¯t.

The output probabilities from the Softmax classification layer are then shown in Ct format. The 26-frame sequence’s extracted deep features are utilized to determine if the series contains aberrant activities or everyday events. This information is sent from LSTM to the SoftMax layer, which makes the final predictions (Ullah et al., 2021). The ideal hyperparameter values were determined through several tests, shown in Fig. 5. We ultimately settled on Adam as the optimizer, and the loss function is categorical cross-entropy with a learning rate of 0.01. For training the model, there were 20–70 epochs. When the loss ceased decreasing, we halted the training process. The equation is applied in the suggested training model for the loss in the RLCNN.

(11) Loss=−1M∑a=1M(∑d=1D∑j=1C+F1{xta=k}lognk).

Figure 5 Work flow for predicting anomalous behavior using proposed model RLCNN.

As previously discussed, CNN is best for image classification, and LSTM is best for working with a data sequence. We have discussed different techniques for image classification and action recognition, but none of those techniques alone could give us a well-suited prediction. Finally, we combined the convolutional neural network and the LSTM network to take advantage of both algorithms and get the required human activities. We will use a convolutional network to extract frames from a video, and then this output will be used in the LSTM network for the action recognition process. Figure 5 explains prediction using the RLCNN model. The LSTM will be responsible for learning temporal information, while the convolutional neural network will be responsible for learning spatial information. The RLCNN approach will merge CNN and LSTM layers into a single model (Poormehdi Ghaemmaghami, 2017). The LSTM model can then use the information gathered by the CNN to predict what will happen in the video. To model the temporal sequence, the LSTM layer(s) get spatial data from the frames from the convolutional layers at each time step. Because the network directly learns spatiotemporal features during training, a robust model, RLCNN, is created in this way. A time-distributed wrapper layer will also be used, allowing us to apply the same layer to each movie frame (Bleed AI Academy, 2023). The dense layer will then forecast the action taken using the results from the LSTM layer with SoftMax activation.

UCF datasets selection criteria

Upon examining UCF50, UCF101, UCFYouTube, and bespoke datasets for identifying anomalies aimed at avoiding theft in mall monitoring, it is evident that all of them capture a plethora of atypical, illicit, and aggressive behaviors recorded in public spaces, such as streets, retail establishments, and educational institutions (UCF Center for Research, 2012). Although UCF50 and UCF101 are significant for broad action recognition, they predominantly comprise brief, edited video segments illustrating sporting or entertainment-related human activities. Likewise, UCFYouTube offers a diverse selection of videos that are beneficial for detecting human behaviors and identifying theft-related irregularities. Consequently, these datasets are optimally aligned with this particular application, emphasizing the various categories of human interactions that facilitate the precise detection of deviant or typical actions in real-world retail store monitoring (UCF Center for Research, 2012).

Preprocessing and collection bias: UCF datasets and anomaly detection

The UCF50, UCF101, and UCFYouTube databases are valuable for generic action recognition; nevertheless, they have several biases that constrain their usefulness to particular scenarios. UCF50 and UCF101 are predominantly derived from web videos that emphasize visually distinct, well-composed actions, frequently executed by persons who overlook routine movements essential for anomaly detection. It fosters a predisposition towards readily identifiable, high-action situations observed in real-world surveillance. Preprocessing procedures, such as segmenting movies into brief snippets, intensify these biases by removing the temporal context crucial for identifying anomalies that develop over extended durations. We want to utilize these datasets for experimental outcomes and will conduct experiments on datasets centered on behaviors in shopping center surveillance cameras in the future.

Datasets

In this research, the RLCNN model was tested using the UCF50, UCF101, UCFYouTube (UCF Center for Research, 2012), and Custom datasets, which contain a lot of unusual, unlawful, and aggressive behavior caught on video in public areas, including streets, stores, and schools. This dataset was chosen because it was compiled from real-world events that might occur anywhere. Additionally, these abnormal behaviors can cause serious issues for individuals and society. A handmade, either a public dataset or one with a particular backdrop and surroundings, is used in several publications (e.g., the dataset from a hockey match and the dataset from a movie), which is uncommon in our daily lives. UCF50 (UCF Center for Research, 2012) is a diversified collection of human actions. It comprises 50 different action classes, where films from each class are grouped into distinct groups that share similar traits, as described in Table 2. For example, one group can contain recordings of someone playing the piano four times, each from a different angle.

Table 2 Dataset.

Dataset	UCF50	UCF101	UCFYouTube	UCFCustomized	
Average videos per action category	100	140	135	145	
Average number of frames per video	199	199	199	199	
Average video frame height	320	320	320	320	
Average video frame width	240	240	240	240	
Average number of frames per video	240	240	240	240	
Average frames per seconds per video	26	26	26	26	

UCF101 (UCF Center for Research, 2012) is a well-known real-world action video dataset that compiles YouTube videos from 101 different action classes, as given in Table 2. There are an equal number of examples from each category, ranging from 100 to 130 samples, and each instance contains an action that lasts between 2 and 7 s. The widest variety of activities is provided by UCF101, which has 13,320 videos across 101 action categories. One of the most challenging datasets to evaluate is the UCFYouTube (UCF Center for Research, 2012). It features four examples, 11 action classes from sports, and movies from 25 genres. The video clips in the same collection have some things in common, such as the same actor, a comparable setting, a comparable point of view, etc. A combination of the UCF101 and UCFYouTube datasets is the UCFCustomized dataset. Both datasets are pooled to provide more training data for the RLCNN model to predict human activities accurately. The RLCNN model will learn more from trained data as the training dataset grows, improving prediction. Longer, straggly security video feeds for many anomalies are included in this dataset, including road accidents, burglaries, explosions, fighting, robberies, and shoplifting, in addition to the category of everyday events. 80% of the data for training and 20% for testing in our trials is used. UCF50, UCF101, UCFYouTube, and UCFCustomized action recognition datasets are used to train the RLCNN model. 20 random categories are chosen. First, a frame of one random video from each genre and the labels that go with it are selected, as given in Table 3.

Table 3 Experimental setup-RLCNN & ConvLSTM parameters.

	Size	
Parameters	RLCNN	ConvLSTM	
Number of filters	16, 32, 64	4, 8, 14, 16	
Kernel size	3 × 3	3 × 3	
Sequence length	20	20	
Number of classes	6	6	
Image width and height	64 × 64	64 × 64	
Activation function	ReLu	Tanh	
MaxPooling	2D	3D	
Recurrent dropout	0.25	0.2	
Optimizer	Adam	Adam	
No. of epochs	70	50	
Platform	Jupiter notebook	Jupiter notebook	
Languages	Python	Python	
Single node system configuration	RAM 8 GB Intel Core i5	RAM 8 GB Intel Core i5	

Performance evaluation metrics

The evaluation measures utilized in this article are defined as follows:

Accuracy: Determines how frequently forecasts and labels match. The metrics generate two local variables, overall and counting, to calculate how often zpredic and ztrue coincide. The binary accuracy of this frequency is ultimately returned by dividing overall by counting (Vosta & Yow, 2022).

Recall: Computed by TP(TP+FN) where TP is True Positive, and FP is False Negative (Vosta & Yow, 2022).

Precision: Computed by TP(TP+FP) where TP is True Positive, and FP is False Positive (Vosta & Yow, 2022).

F1 score: It is a precise and harmonic recall method. The F1 Score is greater as it approaches 1, while 0 denotes the lowest value. The metrics obtained by applying (Vosta & Yow, 2022) are recall and precision: F1=2×(Precision×Recall)(Precision+Recall)

Confusion matrix: Outlines the projected outcomes for the classification problem and displays them in matrix style to indicate the proportion of accurate and inaccurate predictions (Vosta & Yow, 2022). This indicates that it calculates the loss function after each epoch. We must also specify how many frames from each video file we will extract.

The importance of performance evaluation metrics

It is essential to determine which assessment measures are superior for assessing the performance of the suggested model. This study employs generic criteria that yield a more precise assessment of the model. These metrics constitute a standardized framework utilized by the majority of contemporary algorithms to assess their effectiveness. An accuracy provides a broad perspective, although it may be deceptive, owing to the disparity between typical and abnormal activity. Recall gains significance as it measures the system’s capacity to identify all actual theft incidents that directly affect security efficacy. Conversely, accuracy assesses the system’s capacity to reduce incorrect information, hence preserving trust and alleviating the burden on security staff. The F1-score, a harmonic mean of precision and recall, offers a balanced evaluation, essential for developing a system that must concurrently minimize missed thefts and decrease false alarms. The confusion matrix provides a concise overview of the model’s performance, highlighting specific incorrect classifications and facilitating focused enhancements.

Results and discussion

The proposed system uses and evaluates a Jupiter Notebook running Windows 10 with a CoreTM i5 processor and 8 GB of RAM. ConvLSTM, an advanced deep learning technique, is used to excerpt CNN features, LSTM to analyze data sequences, and the RLCNN “classification learner” model to learn action recognition. ConvLSTM, a component of the Keras toolkit, is used to apply the suggested model. To get the best results from our studies, we tuned the model using several hyperparameters, as mentioned in Table 3. The outcomes of our tests are compared in Table 3. As a result, we added more data to our dataset and used Adam as an optimizer in our tests. The model’s learning rate is also set to 10−4. After the loss function converges, the code stops since continuing would be absurd. The epochs are likewise set to 70. The procedure terminates, and the absolute accuracy is the outcome if the difference between the loss functions of the two following epochs is smaller than the tolerance value.

Our model was trained for more classes than usual, and when RLCNN was tested for those classes, the results were encouraging. The results show that the RLCNN model performs very well for selected classes. By using RLCNN, we were able to accomplish the primary goal of the research: to identify human activity in a shopping center to deter theft, shoplifting, and other criminal conduct and help the companies avoid losses. Therefore, we will test the RLCNN model on specific videos. To assess how well our model functions, we compare the outcomes of our experimental work with those from previous approaches used on the UCF50, UCF101, UCFYouTube, and our customized dataset. Different sample datasets evaluate the proposed method’s ability to recognize actions. The RLCNN model used benchmark datasets to produce the best results, satisfying the research objectives. Results for each model are provided in Table 4 and are reviewed in distinct sections, along with comparisons to current best practices. In our trials, we fixed this sequence length to 20. We contrast our suggested approach with a 3D convolutional network by assessing accuracy. ConvLSTM was evaluated, and the computation time increased. Now, we calculated using our hypothetical RLCNN model to obtain a prediction of the recognition of human activities. Working with the RLCNN model, we got the results and compared them with the state-of-the-art models, as is given in Table 5.

Table 4 Performance of RLCNN Model using different datasets.

	Accuracy	
Dataset	Recall	Precision	F1 score	RLCNN	ConvLSTM	
UCF50	0.91	0.90	0.92	0.92	0.80	
UCF101	0.94	0.91	0.90	0.92	0.76	
UCFCustomized	0.93	0.89	0.91	0.96	0.85	
UCFYouTube	0.93	0.89	0.90	0.97	0.91	

Table 5 Comparison of RLCNN with other models.

SCL [72)	Binary SVM (Sandler et al., 2018)	LTR (Lu, Shi & Jia, 2013)	MIL-C3D	TSN (Zhong et al., 2019)	RLCNN [Ours]	
TP	FP	TP	FP	TP	FP	TP	FP	TP	FP	TP	FP	Acc%	
0.15	0.41	0.21	0.22	0.24	0.23	0.38	0.18	0.56	0.16	0.5	0.04	40	
0.37	0.67	0.41	0.37	0.39	0.39	0.58	0.44	0.78	0.5	0.7	0.13	55	
0.43	0.83	0.62	0.63	0.56	0.6	0.64	0.59	0.84	0.66	0.83	0.36	70	
0.68	0.89	0.78	0.79	0.77	0.81	0.77	0.71	0.87	0.81	0.91	0.57	89	
0.89	0.93	0.91	0.90	0.92	0.89	0.83	0.87	0.90	0.91	0.96	0.79	96	

Different models are compared with RLCNN for predictions SCL (Hasan et al., 2016), SVM (Sandler et al., 2018), LTR (Lu, Shi & Jia, 2013), TSN-RGB (Zhong et al., 2019), MIL-C3D (Sandler et al., 2018) with SCL-black, SVM-orange, LTR-blue, MIL-C3D-blue, TSN-RGB-blue, and RLCNN with a purple color scheme. The test classifier displays the likelihood that unusual circumstances will be correctly classified. In that instance, separating the UCF datasets with an excellent accuracy value is challenging since they involve numerous regular activities in situations with real-world variations in lighting and perspectives. The suggested approach performs about as well as C3D. However, the complex dataset structure can necessitate a more challenging feature extraction design. More convolutional layers may be used to access more high-level features. In this instance, we developed a model that used RLCNN rather than ConvLSTM, which improved classification precision.

Table 5 shows how the proposed RLCNN model stacks up against several other top models (SCL, binary SVM, LTR, MIL-C3D, and TSN) at different thresholds. For each model, including the proposed RLCNN, the true positive (TP) and false positive (FP) rates are presented. The RLCNN provides the true positive (TP) and false positive (FP) rates, in addition to an overall accuracy score. Analysis of the experimental results in Table 5 reveals that SCL has a true positive rate of 0.15 and a comparatively elevated false positive rate of 0.41. The binary SVM exhibits a marginally superior true positive rate of 0.21, accompanied by a comparable false positive rate of 0.22. LTR attains a true positive rate of 0.24 alongside a false positive rate of 0.23. MIL-C3D exhibits a superior true positive rate of 0.38, although it also demonstrates an elevated false positive rate of 0.18 in comparison to LTR. TSN demonstrates the highest true positive rate among the evaluated models, at 0.56, accompanied by a false positive rate of 0.16. Conversely, RLCNN in this initial case demonstrates a markedly elevated true positive rate of 0.5 and a minimal false positive rate of 0.04, culminating in an accuracy of 40%.

As we advance through several experimental outcomes, the true positive rates rise, while the false positive rates also generally increase across all models, signifying a trade-off between accurately detecting positive cases and erroneously classifying negative examples as positive. The RLCNN model shows a better balance between high true positive rates and lower false positive rates, often achieving higher accuracy scores across different performance levels shown in the table. In the final scenario, RLCNN attains the greatest true positive rate of 0.96% with a modest false positive rate of 0.79%, resulting in the highest accuracy of 96% among all models at that level. This finding indicates that the RLCNN model may be more proficient in accurately categorizing occurrences while reducing false positives compared to the other models discussed in this comparison. As a result, in our studies, we got somewhat greater accuracy with RLCNN than with ConvLSTM, as shown in Table 6. The primary goal of this research is to effectively discover and recognize anomalies using a combination of CNN, RNN, and LSTM models. CNN models are getting increasingly complex for many computer vision applications, raising concerns about their viability for edge computing. Different models are compared in Table 6.

Table 6 Evaluation of proposed RLCNN model’s performance.

Model	Recall	Precision	F1 score	ACC	
MobileNetv2-LSTM	86	74	77	88	
MobileNetv2-BD-LSTM	79	84	76	87	
MobileNetv2-Res-LSTM	91	78	82	95	
ConvLSTM	78	73	81	85	
RLCNN-Proposed method	79	88	83	96	

Figure 6 illustrates that the confusion matrix reflects the overall efficacy of the proposed RLCNN model in predicting human activities in the UCF datasets. The numbers in the confusion matrix indicate various proportions for multiple instances, demonstrating that the RLCNN model predicted one behavior while the actual activity was another. Figure 6 illustrates that the diagonal values in the matrix denote the proportion of precise forecasts for every action, reflecting the model’s precision for that particular class. The matrix indicates that the model excels at recognizing “pushups” and “diving” actions, as evidenced by their highest percentages. This information can be utilized to identify different human behaviors in diverse locations, such as shopping centers and gatherings. The scores in the confusion matrix indicate that the model failed to predict accurate outcomes for “skiing,” demonstrating reduced accuracy and increased confusion with other activities. The suggested model, RLCNN, achieved an accuracy of 94.23 for pushups and a lesser accuracy of 81.01 for skiing, indicating superior performance compared to previous state-of-the-art approaches.

Figure 6 Human activity prediction using UCF datasets.

Figure 7 presents two graphs: The first graph shows the total loss compared to validation loss and how well the CONVLSTM model performs on the UCF50 dataset, while the second graph shows total accuracy compared to validation accuracy for the same dataset. The initial graph illustrates the loss across 17.5 epochs, with the blue line demonstrating a rapid decline in training losses during the early epochs, indicating the model’s proficiency in learning and reducing errors in the training data. Conversely, the red line denotes the validation loss, which initially declines but begins to fluctuate and increase after approximately 7.5 epochs. Figure 7 indicates that the training loss decreases from approximately 1.75 to below 0.25, while the validation loss stabilizes around 1.0, suggesting a substantial disparity and likely overfitting beyond a certain threshold. The second graph illustrates the loss and validation accuracy across an identical number of epochs and the same dataset employed. The blue line in this graph represents the training accuracy, which consistently rises and attains a value of 1.0. This behavior signifies that the suggested model, CONVLSTM, demonstrates exceptional performance on the training data. Likewise, as illustrated in the second graph of Fig. 7, the red line denotes the validation accuracy, which also ascends; however, it attains a lower value, approximately 0.85, and exhibits variations akin to the validation loss. The experimental results on UCF50 indicate that the disparity in elevated values between training and validation accuracy suggests a likelihood of overfitting. In evaluating values, the training accuracy increases from 0.2 to 1.0, while the verification accuracy ascends from approximately 0.2 to around 0.85. The graphs together indicate that the model effectively learns from the training data.

Figure 7 CONVLSTM loss Vs. accuracy using the UCF50 dataset.

Figure 8 contains two graphs illustrating the training performance of an RLCNN model on the UCF50 dataset throughout 50 epochs. The first graph illustrates total accuracy vs. total validation accuracy, while the second graph depicts total loss vs. total validation loss over the same dataset for an identical number of epochs. The blue line denotes the training accuracy, which consistently rises to 1.0, signifying exceptional performance on the training data. The red line denotes the validation accuracy, which also rises but attains a lower value, approximately 0.9. This outcome exhibits greater variability than the training accuracy. Upon comparison, the training accuracy escalates from approximately 0.2 to nearly 1.0, whereas the validation accuracy ascends from roughly 0.2 to about 0.9. The second segment of Fig. 8 shows the cumulative loss compared to the overall validation loss. The blue line represents the training loss, which decreases rapidly during the initial epochs and continues to decline, nearing zero by the end of training, indicating good performance. The red line denotes the validation loss, which initially declines but then exhibits fluctuations and a modest increase after approximately 30 epochs. Upon comparison of these numbers, it is evident that the training loss decreases from about 1.4 to nearly zero, whereas the validation loss diminishes from roughly 1.4 to about 0.4 before increasing to approximately 0.5. The experimental results indicate that the disparity between the training and validation loss, especially after 30 epochs, reinforces the model’s performance.

Figure 8 RLCNN loss Vs. accuracy using the UCF50 dataset.

We have conducted experiments with several scenarios utilizing diverse datasets with the ConvLSTM model, as illustrated in Fig. 9. We present the training performance of a ConvLSTM model over 17.5 epochs on a UCF-customized dataset. Figure 9 comprises two components: the first depicts total loss vs. total validation loss, while the second illustrates total accuracy vs. total validation accuracy. The initial segment denotes the loss incurred during training and validation. In this experimental section, the blue line, which signifies the training loss, exhibits a rapid decline from about 1.6 to around 0.1, showing efficient learning and error reduction on the training dataset. Conversely, the red line, denoting the validation loss, initially declines but exhibits greater oscillations and ascends to approximately 0.4 after roughly 7.5 epochs. Comparing these numbers reveals that the validation loss markedly exceeds the training loss beyond this juncture, suggesting potential overfitting. The second segment of Fig. 9 illustrates the accuracy during the training and validation phases. The blue line, showing training accuracy, steadily goes up and gets close to 1.0, which means the ConvLSTM model performs very well on the training data. The red line, denoting the validation accuracy, rises but stabilizes at 0.9, displaying fluctuations akin to the validation loss. Relative to other metrics, the training accuracy increases from approximately 0.4 to almost 1.0, but the validation accuracy rises from roughly 0.4 to about 0.9. The disparity between training and validation accuracy, along with the elevated validation loss, underscores the likelihood of overfitting.

Figure 9 CONVLSTM Loss Vs. accuracy using the UCF customized dataset.

We conducted many scenarios with UCF datasets. In this experiment, we implemented our proposed model, RLCNN, as illustrated in Fig. 10. This graphic contains two graphs depicting the training performance of a model across 60 epochs. The first graph shows the overall loss alongside the validation loss, and the second graph shows the total accuracy compared to the validation accuracy for the RLCNN model. In this scenario, the blue line denotes the training loss, which diminishes swiftly during the opening epochs and persists in its decline, ultimately achieving a minimal value of approximately 0.15 by the conclusion of training. The red line denotes the validation loss, which initially declines but exhibits considerable fluctuations, especially after 10 epochs, ultimately concluding at approximately 0.2. We analyzed the resultant numbers and noted that the training loss began at approximately 1.6 and diminished to 0.15, but the validation loss commenced at roughly 1.6 and exhibited fluctuations, concluding at approximately 0.2. The changes in validation loss, especially after 40 epochs, show that the model does better than ConvLSTM; however, it still has slight issues when dealing with new data. When we looked at the overall accuracy and the total validation accuracy, we found good results, showing that the RLCNN model did a great job at identifying different human actions. The blue line illustrates the training accuracy, which escalates swiftly during the initial epochs and approaches 1.0, signifying outstanding performance on the training data. The red line denotes the validation accuracy, which also rises but exhibits greater oscillations and plateaus at a marginally lower value, approximately 0.95. The training accuracy increases from approximately 0.3 to almost 1.0, and the validation accuracy ascends from roughly 0.3 to about 0.95. The disparity between training and validation accuracy, along with the variability in validation accuracy, indicates that the model excels with the training data.

Figure 10 RLCNN loss Vs. accuracy using the UCF customized dataset.

This study has established a model called RLCNN for the recognition of both aberrant and normal behaviors. We have arranged various activities together with their corresponding forecast values. Various activities accompany sequences of procedural frames. The Reality column displays the actual action executed in the sequence, while the Predictions column illustrates the model’s anticipated behavior. We have incorporated the Confidence Score column, which reflects the model’s certainty in its forecast as a percentage. The confidence level signifies the extent to which the model is certain about the prediction aligning with the actual activity value. This number shows that when looking at the different values in the columns, the model is very accurate for activities like YoYo, TaiChi, and Billiards, achieving confidence ratings of 99.50%, 99.00%, and 96.05%, respectively. In these instances, the anticipated actions correspond with the actual outcomes. Nevertheless, the model demonstrates diminished confidence and erroneous predictions for Walking with Dog, misclassifying it as Fighting with Dog with a confidence level of 75.00%. The model indicates uncertainty regarding this behavior, although it predicts that 75% of this action is accurate. In a separate activity, basketball, the model erroneously predicted it as soccer juggling with 56.00% confidence. This indicates that the model excels at recognizing different actions but encounters difficulties with actions that exhibit visual similarity or necessitate nuanced differences. The diminished confidence scores in erroneous predictions reflect the model’s uncertainty in these instances, consistent with the identified mistakes. The table underscores the model’s proficiency in reliably identifying specific actions with considerable confidence while also exposing its shortcomings in distinguishing between visually analogous or intricate activities.

Discussion on visual results

Shopping behavior encapsulates the decision-making processes and actions of individuals who acquire and utilize products. The academic and corporate sectors are focused on analyzing consumer behavior. To guarantee public safety, closed-circuit television (CCTV) surveillance systems are routinely employed to monitor locations with high foot traffic, such as football matches, music festivals, and large establishments like shopping malls. The RLCNN model prioritizes the safety and security of individuals and enterprises in its design. It yields precise outcomes for forecasting human behavior. The RLCNN algorithm is provided with a sample set of videos to accurately predict human behavior. The trials indicate that complex behaviors such as “horse racing” and “horse riding,” which differ primarily in the number of horses involved—many for racing and one for riding—exhibit an accuracy rate of 90%. The dataset’s ground truth and projected classes exhibit comparable visual content. For example, “skiing” is anticipated to be categorized as “skateboarding,” while “diving” is incorrectly designated as “high jumping.” Nonetheless, these problematic classes have comparatively low confidence scores for erroneous predictions. The assertion “high jump” is inaccurate, as the diver’s leap throughout the dive exemplifies the high jump category for that specific time frame in the footage. Prior models necessitated extended computation durations, whereas RLCNN trained rapidly and generated activity predictions promptly. This method yields optimal accuracy and is straightforward to master. Consequently, the RLCNN model enhances performance relative to the leading models, as demonstrated in Table 6. We were able to identify only one individual in the frame. To accurately assess an individual’s behaviors, just one person must be present in the frame at any given time. We were unable to create a system that utilizes several databases and UCF datasets to identify activities with numerous participants. We were unable to establish an action recognition system utilizing multi-view surveillance movies integrated into a visual sensor network across several dynamic situations. However, future development would resolve all these limitations.

Discussion

This research conducted numerous experiments on the base model CONVLSTM and the suggested model RLCNN using UCF datasets for both models. The experimental findings indicate that our suggested model, RLCNN, outperforms the baseline model, CONVLSTM, regarding accuracy and loss metrics. The various UCF datasets were utilized to train the model and subsequently tested on multiple human activities. The study achieved superior performance in predicting several actions that indicate whether the activity is normal or pathological. This study employed deep learning models, including CNN for feature extraction and LSTM for identifying anomalous sequences, to detect unusual human behavior in crowded shopping malls. Table 4 indicates that this study achieved accuracy for each dataset employed to evaluate the RLCNN model using four UCF datasets. Our suggested model attained an accuracy of 97% on the UCFYouTube dataset and 96% on the UCFCustomized dataset, demonstrating superior performance in identifying abnormal behavior in crowds.

The experimental results are presented in Figs. 7–10, which compare the total training loss and accuracy with the total validation loss and accuracy derived from UCF datasets. This research utilized four UCF datasets for training and testing objectives. This study assessed the efficacy of the suggested model in comparison to state-of-the-art approaches, as presented in Table 5. Figures 7–10 illustrate the training graph, loss, and class-wise accuracy of the proposed model. The proposed method predicted accurate and erroneous predictions generated for a singular action movie, together with their highest confidence scores. A collection of frames for each action is also supplied for the readers’ use. It illustrates the visual results of our proposed technique for detecting anomalous activity and demonstrates the accurate and faulty prediction outcomes of the recommended model.

For evaluation purposes, we supplied a sample collection of movies depicting various activities to our model RLCNN to anticipate human behavior. The tests indicate that the complex actions of “horse racing” and “horse riding,” which differ only by the number of horses—many for racing and one for riding—achieve an accuracy rate of 90%, which is impressive given the difficulty of these situations. The dataset’s ground truth and projected classes display similar visual content. We note that prior models required prolonged calculation times, whereas RLCNN trained swiftly and produced activity predictions expeditiously. This technique produces maximum precision and is easy to learn. Thus, the RLCNN model improves performance compared to the top models, as illustrated in Table 6. We could identify only one individual in the frame. To precisely evaluate an individual’s behaviors, just one person needs to be present in the frame at any moment. We could not develop an action recognition system using multi-view surveillance movies incorporated into a visual sensor network across many dynamic scenarios. Nonetheless, further advancements will address all these constraints.

Conclusion

The RLCNN model is adept at anomaly detection in shopping center surveillance films, as it proficiently captures the video data’s temporal and spatial dimensions. Surveillance recordings are intrinsically sequential, depicting events that occur across time. The recurrent connections in the RLCNN let it retain knowledge from previous frames, facilitating comprehension of the present frame and forecasting future events. This is essential for identifying anomalies that entail behavioral changes over time, such as loitering or hiding goods. Moreover, the convolutional layers of the RLCNN effectively extract local spatial data from each frame, including object movements and interactions. The RLCNN synergizes the advantages of RNNs and CNNs in proficiently discerning intricate patterns of typical activity and detecting anomalies that may signify theft. We have developed a model class called RLCNN that is flexible enough for various vision problems that require sequential inputs and outputs. According to our research, learning sequential dynamics using a deep sequence model can outperform learning only a deep hierarchy of visual parameters or learning only the dynamics of the output sequence using a fixed visual representation of the input. Deep sequence modeling methods like RLCNN are becoming more and more critical to vision systems for issues with sequential structure as the area of computer vision develops beyond jobs with static input and predictions. These techniques are a good solution for perceptual cases involving time-varying visual input or sequential outputs since they integrate easily into current optical recognition pipelines and require little to no hand-designed features or input preprocessing. Our model is not capable of handling several people engaged in various activities. Only one other person should be in the frame to appropriately identify that person’s actions. This is because we had to deal with data to train our model, and we had data in that shape. These restrictions would be lifted in the future. Various databases and UCF datasets will be utilized to identify activities that involve many people. Although it is a very computationally expensive procedure, we will continue to use it in the future with various setups and parameters.

Limitations

The application of UCF datasets (UCF50, UCF101, and UCFYouTube) for the detection of anomalous actions, especially in contexts such as commercial store monitoring, entails several notable constraints: The existing RLCNN approach is limited to efficiently analyzing actions executed by a solitary individual within the frame. It cannot effectively manage situations with several individuals participating in various activities concurrently.

The model’s existing capabilities are significantly dependent on the particular organization of the training data.

The RLCNN paradigm is resource-intensive, potentially restricting its use in real-time applications or on devices with constrained processing capabilities.

These datasets fail to encompass extended, continuous video streams and the diverse lighting conditions characteristic of real-world surveillance complexities.

UCF datasets concentrate on human activities such as sports, exercise, and entertainment; however, they are deficient in identifying prolonged behavioral anomalies linked to theft.

The present study has utilized UCF datasets exclusively for abnormal behavior detection, thereby constraining the model to specific tasks.

Future work

Considering the constraints of current datasets such as UCF for the detection of anomalous activity in shopping mall surveillance, we propose many avenues for further research and development: Obtain and organize datasets that encompass situations featuring several individuals engaging in various activities.

Implement advanced object-tracking techniques to identify and monitor individual subjects within the scene.

Assess the model’s efficacy on demanding real-world datasets, particularly those obtained in uncontrolled settings.

Design real-time systems: Enhance the model for real-time applications, including video surveillance, human-computer interaction, and assistive technology.

In the future, we will develop specialized datasets that precisely represent real-world difficulties in detecting anomalous activity for shopping mall surveillance.

We will investigate synthetic data generation utilizing realistic virtual surroundings and simulated activities to enhance scarce real-world data.

In the future, we will utilize sophisticated anomaly detection methodologies, like graph neural networks and temporal modeling with LSTMs or transformers, to identify subtle anomalies and temporal relationships.

Real-world impact: applications of anomaly detection in retail security

The findings of a study on anomaly identification for avoiding theft in shopping mall monitoring had substantial practical implications, transcending mere reduction of stealing occurrences. Reinforced operational security: This methodology can be utilized to safeguard assets by examining patterns of abnormal activity, enabling security professionals to identify high-risk places and periods.

Commercial theft mitigation: Mitigating theft directly results in diminished revenue losses for retailers. This could end up in higher earnings for enterprises.

Enhanced client satisfaction: By reducing unpleasant crime occurrences, stores can foster a more enjoyable and safe purchasing environment.

Applications for public safety: The ideas of recognizing anomalies in monitoring can be used in several security contexts, including the identification of suspicious activity in transit systems, public areas, and event settings.

Intelligent building administration: The methods can be incorporated into comprehensive intelligent building administration systems, enhancing general safety and effectiveness.

Ethical considerations for human data in technical research

The UCF dataset is accessible to the public and is frequently utilized for research endeavors. The UCF dataset employed in this work raises no privacy concerns. This dataset is open-source and accessible to the public for research purposes. This study has downloaded the entire UCF dataset to perform the experiments and has saved it on a local PC. This dataset was stored on a secure drive to prevent any data interference. We adhered to ethical principles and procedures in utilizing publicly accessible computer vision datasets. No ethics review board permission was necessary as the dataset was publicly accessible online; nonetheless, we did contemplate the ethical implications of using this dataset. This study ensured that neither humans nor animals suffered harm nor participated in any illegal activities.

Data availability

All data supporting this study are openly available within the article and online (UCF Center for Research, 2012) as open source without restrictions. The DOI to access the code repository can be found at https://doi.org/10.5281/zenodo.15243236.

Supplemental Information

Supplemental Information 1 Code and UCF 50 Dataset.

Supplemental Information 2 Code and UCF 10 Dataset.

Supplemental Information 3 Code and Customized UCF Dataset.

Supplemental Information 4 Code and UCF Youtube Dataset.

Supplemental Information 5 Readme.

Additional Information and Declarations

Competing Interests

The authors declare that they have no competing interests.

Author Contributions

Muhammad Sajid conceived and designed the experiments, performed the experiments, analyzed the data, performed the computation work, prepared figures and/or tables, authored or reviewed drafts of the article, and approved the final draft.

Ali Haider Khan conceived and designed the experiments, performed the experiments, performed the computation work, prepared figures and/or tables, authored or reviewed drafts of the article, and approved the final draft.

Kaleem Razzaq Malik conceived and designed the experiments, performed the experiments, analyzed the data, prepared figures and/or tables, authored or reviewed drafts of the article, and approved the final draft.

Javed Ali Khan conceived and designed the experiments, performed the experiments, analyzed the data, performed the computation work, authored or reviewed drafts of the article, and approved the final draft.

Ayed Alwadain analyzed the data, prepared figures and/or tables, and approved the final draft.

Data Availability

The following information was supplied regarding data availability:

The data is available in the Supplemental Files.

The implemented code (code configuration files, source code, and a user manual) used in this study is available at GitHub and Zenodo:

- https://github.com/SajidAuK/Anomaly-Detection-in-Shopping-Center.

- Muhammad Sajid, & Muhammad Sajid. (2025). auksajid/Anomaly-Detection-in-Shopping-Center: Anomaly Detectopm (v1.0.0). Zenodo. https://doi.org/10.5281/zenodo.15243236.

The UCF101 - Action Recognition Data Set is available at https://www.crcv.ucf.edu/data/UCF101.php.

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
