# Peer review of "A new approach of anomaly detection in shopping center surveillance videos for theft prevention based on RLCNN model"

_PeerJ Computer Science, doi:10.7717/peerj-cs.2944_

## Round 0.1 · original submission · Major Revisions

Dear Authors,

Thank you for submitting your manuscript entitled "A new approach of anomaly detection in shopping center surveillance videos for theft prevention based on RLCNN model" to our journal.

After a thorough review process, we have received feedback from three expert reviewers. While all agree that your work addresses an important and timely problem with promising methodological innovations, there are several critical aspects that require substantial revision before the manuscript can be considered for publication.

Accordingly, the editorial decision is: Major Revision.

The reviewers have pointed out the need to:

- Improve the clarity and organization of several sections, especially the introduction, related work, and methodology.
- Expand and update the literature review, ideally including a comparative table summarizing key prior works.
- Enhance the visual quality of the manuscript, including clearer figures, and add flowcharts or diagrams where appropriate to better illustrate your approach.
- Provide more detailed explanations of the datasets, preprocessing steps, evaluation metrics, and experimental setup.
- Include a deeper discussion of the study’s limitations and future research directions.

We believe your work has potential for publication pending these substantial improvements. We encourage you to carefully address each comment from the reviewers and submit a point-by-point response along with the revised manuscript.

We look forward to your resubmission.

**Language Note:** PeerJ staff have identified that the English language needs to be improved. When you prepare your next revision, please either (i) have a colleague who is proficient in English and familiar with the subject matter review your manuscript, or (ii) contact a professional editing service to review your manuscript. PeerJ can provide language editing services - you can contact us at [email protected] for pricing (be sure to provide your manuscript number and title). – PeerJ Staff

Reviewer 1 ·

Basic reporting

The article is written in clear and professional English. The introduction provides adequate background and context for the study, ensuring that the reader understands the motivation behind the research. The literature is well referenced and relevant, demonstrating a thorough review of prior work in the field.
The structure of the article conforms to PeerJ standards and follows disciplinary norms. The formal results include clear definitions of terms and theorems where necessary. However, additional clarity on some technical terms and a more explicit articulation of the research problem could improve the introduction.

suggestions for improvement:
1. Expand the introduction to provide a more detailed discussion of the research gap.
2. Ensure all technical terms are clearly defined for a broader audience.
3. Strengthen the literature review by incorporating additional recent works, if available.

Experimental design

The study falls within the aims and scope of the journal. The investigation is conducted rigorously, adhering to high technical and ethical standards. The methods are described in detail, allowing for replication. The paper provides adequate details about the data preprocessing steps. The evaluation methods, assessment metrics, and model selection processes are well described. The sources used are appropriately cited, with proper paraphrasing and quotations.

Suggestions for Improvement:
1. Provide more details on dataset selection criteria.
2. Clarify any potential biases in data collection and preprocessing.
3. Include additional justifications for the choice of evaluation metrics, if necessary.

Validity of the findings

The findings are well supported by the experimental results. The authors provide meaningful conclusions that are aligned with the study’s objectives. The discussion appropriately addresses the results, their significance, and their implications. The paper does a good job of situating the findings within the existing literature. However, it would benefit from a more explicit discussion of any limitations and future directions.

Suggestions for Improvement:
1. Expand the discussion on the limitations of the study.
2. Clearly outline future research directions based on the findings.
3. Discuss potential real-world applications of the study’s results.

Additional comments

Overall, the article is well-structured and meets the journal’s standards. With minor revisions to clarify methodology, enhance literature discussion, and expand on limitations, the paper would be significantly strengthened.

Reviewer 2 ·

Basic reporting

- The language used throughout the manuscript is clear, understandable and professional. Technical terms and explanations are used appropriately, but some sections could be more clearly defined.
- In order to make the literature review more readable and understandable, it would be useful to add a comparative table. This table could include the purpose of each previous study, the method used, the analyses and the results obtained.
- The structure generally seems to be in line with PeerJ standards. The study is organized according to accepted disciplinary norms. However, some sections could be better organized. It is recommended that the "Related works" section be presented as a separate section. In this way, the reviews of previous studies can be systematically discussed and the reader can have a clearer perspective on the subject. In addition, a clearer distinction can be made in the results and discussion sections.
- The introduction section clearly introduces the purpose and motivation of the research. However, the motivation section can be discussed in more depth. In particular, the innovative approach that is intended to be presented in the study and why this study is important can be emphasized more.
- In the results section, terms are clearly defined and explanations are made regarding the performance of the models used. However, some technical terms and concepts could be defined in more detail. Furthermore, the results may need to be substantiated in more detail, particularly the methods used and the discussions on their success could be strengthened.

Experimental design

- The manuscript focuses on a topic that is appropriate for the journal's purpose and scope. However, it may be necessary to discuss more specifically the extent to which the study topic falls within the scope of the journal, especially specific applications related to security and human movement.
- The manuscript has been technically detailed, but more information on ethical standards may be needed. Ethical issues, especially the use of human data, privacy and security, may be addressed more.
- Although the experimental processes are explained in sufficient detail in the methods section, it can be said that more details should be given on the software, code and infrastructure required to repeat the experiments.
- There is limited discussion of the data preprocessing process, but each stage of this process may need to be addressed in more detail.
- A clearer explanation of the evaluation metrics used and how these metrics are calculated for each model may increase the comprehensibility of the methodology.
- Although the sources are generally cited appropriately, some older studies are referenced, and it is possible to consider more recent developments.

Validity of the findings

- A more in-depth discussion of the innovative aspects of the study and its potential impact on the field would be useful, and it would be more explicit in stating why this research is important and how it will contribute to the development of the field.
- Although the results are correctly stated to support the findings, some generalizations could be made more carefully. More focus could be placed on the consistency of the results with the hypotheses of the study.
- A clearer explanation of each of the experiments, data sets used, and tests conducted could increase the reliability of the study. In addition, a more extensive discussion of the tests and results conducted under different scenarios could be useful.
- Although the discussions and arguments regarding the goals set in the introduction section are somewhat strong, the process of achieving the goals is sometimes weaker. It could be better explained how the research progressed step by step and what results were obtained at each stage.
- Although some limitations and future studies were briefly mentioned in the results section, a more in-depth discussion could be made. A more detailed assessment of the limitations of the study and possible directions for future research may be required.

Additional comments

The manuscript addresses an important issue and offers a solution proposal. However, the deficiencies are quite obvious and the technical contributions and methods used need to be clearly stated. Resolving the deficiencies will make the study stronger in terms of scientific and technical aspects.

Introduction:
- "Nonetheless, identical conduct may be anticipated and deemed pathological under realistic conditions" (lines 87-88). The meaning of the sentence is a bit unclear. It could be clearer as "Identical conduct may be expected and yet considered abnormal depending on the specific scenario".
- The “Primary Contributions” section does not detail exactly in which aspects the proposed model offers improvements.

Background:
- Although the section mentions some previous studies, it does not detail to what extent the proposed method is better than other methods.
- It is suggested to add a comparative table including the purpose, method, analysis and findings in order to make the literature review more readable and understandable.

Methodology:
- Some terms and concepts used in the text are not sufficiently explained. For example, technical terms such as “RLCNN” or “ConvLSTM” should be introduced in more depth and additional information should be provided to help the reader fully understand these concepts.
- Some references in the text (e.g., [37], [39], [51]) are used quite frequently and intensively, but the context of these references is not fully provided. The sources may need to be explained in more detail to suit the context.
- Theoretical explanations of the methods and algorithms are quite dense, but little information is given about real-world applications or examples.
- A general flow chart with a strong visual aspect should also be included in the methodology section to better understand the technical process for the reader.
- It is observed that the images in the manuscript are of low quality. This may make it difficult for the reader to understand the technical process and reduce the impact of the images.

Results and Discussion:
- The results in Table 4, where the RLCNN model is compared with other methods, are presented only as a superficial comparison, rather than a more in-depth discussion. "motivations" are mentioned, but how these two factors compete with each other or mix in the real world is not discussed.
- In the manuscript, it is stated that there are confusions between classes such as "diving" and "high jumping", but the reasons for this situation and solutions are not discussed.
- More details should be provided on technical issues such as "multiple layers" or "multiple layers", and solution suggestions should be explained more concretely.
- Performance comparisons and results should be discussed in more depth, and which factors affect these results should be explained clearly. Such superficial expressions make it difficult for the reader to understand the complexity of the technical process and reduce the chance of evaluating the real success of the model.

One of the biggest shortcomings is the inadequacy of visual and figure-based explanations. When figures and flow charts are missing, it becomes difficult for the reader to understand the proposed method and the clarity of the study is negatively affected. Therefore, the manuscript had to be revised and made visually stronger and more descriptive.

---

## Round 0.2 · accepted · Accept

The authors have addressed all of the reviewers' comments. Congratulations!

Reviewer 2 ·

Basic reporting

The revised manuscript is written in clear, professional English, with a well-structured introduction and relevant, well-cited literature. It adheres to PeerJ standards, clearly conveys the study’s motivation, and presents formal results with precise definitions and appropriate proofs.

Experimental design

The revised manuscript falls within the journal’s Aim and Scope and article type. It reflects a rigorous investigation conducted to high technical and ethical standards. The methods are described in sufficient detail for replication, including code, datasets, and computational setup. Data preprocessing is adequately discussed, and evaluation metrics, methods, and model selection are clearly explained. References are appropriately cited and properly paraphrased where necessary.

Validity of the findings

The manuscript presents clearly stated and well-supported results, encouraging meaningful replication through a logical framework and clear relevance to the field. Experiments and evaluations are satisfactorily conducted, and the argument aligns well with the objectives outlined in the introduction. The conclusion effectively identifies open questions, limitations, and directions for future work.

Additional comments

The revision has been largely accomplished, and the manuscript has reached an acceptable standard for publication.